# Cardioprotection by SGLT2 Inhibitors—Does It All Come Down to Na^+^?

**DOI:** 10.3390/ijms22157976

**Published:** 2021-07-26

**Authors:** Maximilian Trum, Johannes Riechel, Stefan Wagner

**Affiliations:** Department of Internal Medicine II, University Hospital Regensburg, 93053 Regensburg, Germany; maximilian.trum@ukr.de (M.T.); johannes.riechel@stud.uni-regensburg.de (J.R.)

**Keywords:** SGLT2 inhibitor, heart failure, HFrEF, HFpEF, arrhythmia, Na^+^, NHE1, late I_Na_, CaMKII

## Abstract

Sodium-glucose co-transporter 2 inhibitors (SGLT2i) are emerging as a new treatment strategy for heart failure with reduced ejection fraction (HFrEF) and—depending on the wistfully awaited results of two clinical trials (DELIVER and EMPEROR-Preserved)—may be the first drug class to improve cardiovascular outcomes in patients suffering from heart failure with preserved ejection fraction (HFpEF). Proposed mechanisms of action of this class of drugs are diverse and include metabolic and hemodynamic effects as well as effects on inflammation, neurohumoral activation, and intracellular ion homeostasis. In this review we focus on the growing body of evidence for SGLT2i-mediated effects on cardiac intracellular Na^+^ as an upstream mechanism. Therefore, we will first give a short overview of physiological cardiomyocyte Na^+^ handling and its deterioration in heart failure. On this basis we discuss the salutary effects of SGLT2i on Na^+^ homeostasis by influencing NHE1 activity, late I_Na_ as well as CaMKII activity. Finally, we highlight the potential relevance of these effects for systolic and diastolic dysfunction as well as arrhythmogenesis.

## 1. Introduction

SGLT2 inhibitors (SGLT2i) were designed as antidiabetic drugs lowering blood glucose levels by selective inhibition of the sodium-glucose cotransporter 2 (SGLT2) in the renal proximal tubule with consequent increase in glycosuria [1]. Although tight glycemic control, which can be achieved by the addition of SGLT2i to the antidiabetic treatment regimen, has the potential to positively affect major adverse cardiovascular events (MACE) [2,3], the risk of myocardial infarction or stroke was not significantly reduced by treatment with SGLT2i [4,5,6]. Moreover, a mediation analysis of EMPA-REG OUTCOME by Inzucchi et al. demonstrated that the reduction in HbA1c levels by empagliflozin contributed only slightly to the observed cardioprotective effects [7]. These data are corroborated by recent studies in patients with heart failure, in whom SGLT2i prevented hospitalization for heart failure and cardiovascular death in diabetic and nondiabetic patients alike [8,9]. Furthermore, a direct SGLT2-dependent effect on cardiomyocytes can be excluded due to the lack of SGLT2 expression in the heart [10,11,12]. Therefore, many (pre-) clinical trials were performed to gain further mechanistic insights. Consequently, a myriad of potential mechanisms was proposed including direct cardiac effects such as inhibition of cardiac Na^+^/H^+^ exchanger 1 (NHE1) [13,14,15], Ca^2+^/calmodulin-dependent protein kinase II (CaMKII) [10,16], and late Na^+^ current (late I_Na_) [17]. Intriguingly, these proteins are centrally involved in cardiac Na^+^ homeostasis, which is fundamentally disrupted in heart failure [18,19,20], making it an interesting target for heart failure treatment. 

In this narrative review we shortly summarize the current knowledge on Na^+^ homeostasis in heart failure and elaborate on potential pathways by which SGLT2i could improve cellular Na^+^ handling and thus prevent heart failure progression.

## 2. Cardiac Na^+^ Handling

### 2.1. Na^+^ Handling in the Healthy Heart

In cardiomyocytes Na^+^ homeostasis is finely tuned by an orchestrated activity of several transporters, ion channels and kinases reflecting its great importance for fundamental cellular processes as excitation-contraction coupling and mitochondrial metabolism. 

During the cardiac action potential (AP), activation of voltage-gated Na^+^ channels (Na_v_1.5) mediates the upstroke (phase 0) of the cardiac AP and thus allows for the activation of voltage-gated L-type Ca^2+^ channels (LTCC), leading to Ca^2+^-induced Ca^2+^ release (CICR) from the sarcoplasmic reticulum (SR) and finally activation of the contractile apparatus. Since voltage-gated Na^+^ channels undergo rapid inactivation (few ms) the magnitude of Na^+^ influx mediated by peak Na^+^ current (peak I_Na_) is small. However, beside peak I_Na_, a small persistent I_Na_ component has been described that is generated by Na^+^ channels exhibiting a sustained bursting activity even at negative membrane potentials [21]. In addition, during the plateau phase of the cardiac action potential, Na^+^ current steady-state inactivation and activation can overlap resulting in the generation of a long-lasting I_Na_ window current. Both latter Na^+^ current components (together labelled as late I_Na_) have a low amplitude, but their long duration leads to a significant amount of Na^+^ entry into the cell [20,22,23,24,25].

Furthermore, Na^+^ influx can be mediated by the cardiac isoform of the Na^+^/H^+^ exchanger (NHE1), which couples the extrusion of one H^+^ with the influx of one Na^+^ and is thus also centrally involved in the regulation of intracellular pH (pH_i_) [26]. In addition, Na^+^ homeostasis is tightly coupled with cellular Ca^2+^ homeostasis via the electrogenic sarcolemmal Na^2+^/Ca^2+^ exchanger (NCX), which exchanges 3 Na^+^ for 1 Ca^2+^ and constitutes—together with SR Ca^2+^ ATPase 2a (SERCA2a)—the main means of cytosolic Ca^2+^ removal during diastole [27,28]. Indeed, in the healthy heart, NCX is responsible for the extrusion of all Ca^2+^ that entered the cell via LTCC in order to maintain a steady-state condition [28,29]. Importantly, NCX can operate in a bidirectional fashion depending on ion gradients across the membrane as well as membrane potential. In “forward mode,” which is favored by increased [Ca^2+^]_i_ and membrane potentials negative to the Nernst potential of NCX, Ca^2+^ is removed from the cell, as mentioned above, while Na^+^ enters the cell. The opposite is true for NCX “reverse mode,” which is promoted by an increase in [Na^+^]_i_ and positive membrane potentials. Apart from these transporters, which are accountable for most of the Na^+^-entry (NCX > Na_v_ > NHE) [28,30,31] further Na^+^ influx is mediated by Na^+^/HCO_3_^−^ cotransporter, Na^+^/K^+^/2Cl^−^ cotransporter, as well as Na^+^/Mg^2+^ exchanger [31,32]. Importantly, while SGLT2 is not expressed in the heart, SGLT1 expression was repeatedly confirmed in healthy hearts and is even upregulated in cardiac disease potentially contributing to Na^+^ influx [33,34]. 

The inward gradient for Na^+^ is maintained by the energy consuming sarcolemmal Na^+^/K^+^-ATPase transporting 3 Na^+^ out of the cell in exchange with 2 K^+^ and simultaneously building up the negative resting membrane potential, which is primarily dictated by the K^+^ conductivity of the cell membrane.

Besides its importance in cardiac excitation-contraction coupling, [Na]_i_ is also involved in cellular energy supply as well as mitochondrial redox regulation, as recently reviewed by Bertero et al. [35]. In brief, mitochondrial Ca^2+^ concentration ([Ca^2+^]_m_), which is elevated in parallel to cytosolic Ca^2+^ levels, e.g., during β-adrenergic stimulation, activates Krebs cycle dehydrogenases to increase the production of reducing equivalents (NADH and NADPH) in order to meet the energetic demand and also to preserve the anti-oxidative capacity of the cell [36]. While mitochondrial Ca^2+^ uptake is mediated via the mitochondrial Ca^2+^ uniporter (MCU) [37], activity of the mitochondrial Na^+^/Ca^2+^ exchanger (NCLX) primarily causes Ca^2+^ extrusion linking [Na]_i_ with [Ca^2+^]_m_ [28,38]. Consequently, an increase in [Na]_i_ results in a decrease in [Ca^2+^]_m_, which in turn hampers Krebs cycle activity [36].

Given the role of Na^+^ homeostasis for cardiac excitation-contraction coupling as well as cardiac metabolism it comes as no surprise that dysregulation of Na^+^ handling is centrally involved in the development and progression of heart failure.

### 2.2. Na^+^ Dysregulation in Heart Failure 

Heart failure is characterized by cellular Na^+^ overload as a consequence of a dysbalance between Na^+^ in- and efflux. Indeed, bulk cytosolic [Na^+^]_i_ from failing compared to non-failing myocytes is elevated by ~2 to 6 mM [18,24,31,39,40,41]. An increase in Na^+^ influx is predominantly caused by (1) an increase in late Na^+^ current (late I_Na_), (2) an increase in NHE1 activity and (3) increased NCX forward mode due to cytosolic Ca^2+^ overload in failing cardiomyocytes [31] (Figure 1).

#### 2.2.1. Late I_Na_

As stated above, the vast majority of voltage-gated Na^+^ channels inactivate rapidly leading to a marked membrane depolarization but only little Na^+^ influx. In contrast, there is a small proportion of channels, which remain activated or close and quickly reopen, mediating a sustained Na^+^ current termed late I_Na_ [20,22,23,24,25,42]. The molecular mechanisms for this peculiar gating behavior are not fully understood. Interestingly, in the recent years it has become evident that beside Na_v_1.5 also neuronal Na_v_ isoforms (e.g., Na_v_ 1.8) [43,44,45,46] are involved in the generation of late I_Na_. Even though this current is small under physiological conditions (~0.5% of peak I_Na_), it is upregulated in heart failure, where it consequently gains further impact on AP duration and is a major source of cytosolic Na^+^ overload [19,20,23,25,47]. Several mechanisms might account for the increase in late I_Na_ in the failing heart including hypoxia [48], mechanical [49] and oxidative stress [19,23,25]. In this context, Ca^2+^/Calmodulin-dependent kinase IIδ (CaMKIIδ), a serine/threonine kinase that is markedly upregulated in heart failure and involved in heart failure development and progression [50,51,52], occupies a central role. In the recent years, a plethora of evidence has accumulated for CaMKII-dependent stimulation of late I_Na_ in a variety of cardiac diseases [24,47,53]. For example, augmentation of late I_Na_ by reactive oxygen species (ROS) with consequent disturbances of Na^+^ and Ca^2+^ homeostasis requires oxidative activation of CaMKII [25,54]. In accordance, three CaMKII phosphorylation sites were detected at the Na_v_1.5 I-II linker loop (S516, S571 and T594) [55,56]. Interestingly, the reported shift of I_Na_ steady state inactivation (SSI) to more negative potentials and increase in late I_Na_ induced by CaMKII-dependent Na_v_1.5 phosphorylation [24] could only be reproduced by phosphorylation of S571 [56], while phosphorylation of S516 and T594 resulted in a leftward shift of SSI without increasing late I_Na_ [55]. To add further complexity, Na^+^ channel β-subunit composition might additionally be relevant for the regulation of late I_Na_ [57]. In conclusion, while the relevance of specific phosphorylation sites is still under debate, there is consensus that CaMKII-dependent Na^+^ channel phosphorylation is the major driver for late I_Na_ augmentation in heart failure. However, although inhibition of late I_Na_ e.g., by ranolazine has been shown to significantly decrease [Na^+^]_i_ in human heart failure [19], this current alone cannot account for the observed Na^+^ overload in failing cardiomyocytes [24,58]. 

#### 2.2.2. NHE1

Indeed, several studies demonstrated an upregulation of NHE1 activity in heart failure [59,60]. In a rabbit model of combined pressure and volume overload induced heart failure, Baartscheer and colleagues showed an increased Na^+^ influx, which could be blunted by inhibition of the increased NHE1 activity with the specific inhibitor cariporide [60]. Interestingly, cariporide could also partly rescue the disturbed Ca^2+^ handling in these myocytes [60]. In a follow-up experiment using the same rabbit model, this working group was also able to show that chronic cariporide treatment (for 3 months after operation) prevented heart failure development by alleviating structural and cellular remodeling [61]. It is still unclear, if the observed increase of NHE1 activity in heart failure is due to an increase in channel expression, posttranslational modification of the channel, or both. We have recently reported that NHE1 expression is markedly upregulated in ventricular tissue of patients with end-stage heart failure as compared to patients with left ventricular hypertrophy and normal systolic function [15]. Intriguingly, NHE1 expression has also been shown to be upregulated in the failing right ventricles of a rat model of pulmonary hypertension [62]. Furthermore, treatment of neonatal rat ventricular cardiomyocytes with aldosterone, a known driver of cardiac hypertrophy and heart failure, results in an increased NHE1 expression with consequent stimulation of hypertrophic signaling, which was ameliorated by cariporide treatment [63]. On the other hand, Yokoyama et al. did not observe any alteration in cardiomyocyte NHE1 expression in a small cohort of patients with chronic end-stage heart failure undergoing heart transplantation compared to non-used donor hearts, despite a markedly increased NHE activity in the diseased hearts [59] pointing to a relevance of post-translational modification. Importantly, NHE1 has been shown to be readily phosphorylated by several kinases—including CaMKII—resulting in a stimulation of channel function [64,65]. 

#### 2.2.3. Na^+^/K^+^ ATPase 

As stated above, Na^+^ overload can also result from a reduced capacity to extrude Na^+^ from the cytosol, a process, which is mainly mediated by the Na^+^/K^+^ ATPase. Several studies have shown a decrease in the expression level of the Na^+^/K^+^ ATPase in animal models of heart failure as well as human heart failure [66,67,68]. Furthermore there might be a shift in the composition of different isoforms of the α subunit (α1, α2, α3), which show different affinities to Na^+^, as well as the β subunits (β1 and β2, with β1 being the prevailing isoform in the heart) [39,67,68,69,70]. While these subunit shifts could result in different activity and affinity of the pump, the available data are inconsistent maybe due to species differences or differences in the etiology or stage of heart failure. The expression level and phosphorylation of phospholemman (PLM), the main regulator of Na^+^/K^+^ ATPase activity, by protein kinases A and C (PKA, PKC) adds a further layer of complexity, especially as data are again conflicting. In a rabbit model of volume-overload-induced heart failure PLM appeared to be hyperphosphorylated [69], while downregulation of inhibitor-1 (I-1) with consequent increase in protein phosphatase-1 (PP-1) activity resulted in PLM hypophosphorylation in human heart failure [71]. Intriguingly, several studies failed to observe a reduction in Na^+^/K^+^ ATPase activity in the failing heart [18,60] making the contribution of this enzyme to Na^+^ overload still a matter of debate.

#### 2.2.4. NCX

In contrast to the Na^+^/K^+^ ATPase the relevance of NCX is better characterized. NCX has a special role in ion dysbalance in heart failure as its activity is tightly coupled to [Na^+^]_i_ and [Ca^2+^]_i_, as described above (see Section 2.1). Importantly, a majority of studies have shown that NCX expression and activity are upregulated in heart failure [72,73,74] with potentially far-reaching consequences.

On the one hand, an increase in [Na^+^]_i_ may induce NCX reverse mode in the early phase of the action potential and hamper Ca^2+^ extrusion via NCX forward mode later on potentially contributing to SR Ca^2+^ loading, which might in turn mitigate contractile dysfunction [75,76]. On the other hand slowing of diastolic Ca^2+^ removal from the cytosol contributes to diastolic dysfunction [77,78]. However, during the cardiac cycle NCX primarily operates in its forward mode, which in combination with a decreased SERCA activity and an increased SR Ca^2+^ leak unloads the SR resulting in decreased Ca^2+^ transient amplitude [72,79,80,81]. At the same time the NCX-mediated Na^+^-influx generates a depolarizing transient inward current (I_ti_) potentially causing cellular pro-arrhythmic events in form of delayed afterdepolarizations (DAD) [82,83]. Moreover, the increase in [Na^+^]_i_ indirectly reduces the availability of reducing equivalents, which not only leads to a deficit in ATP but also contributes to oxidative stress (as outlined in Section 2.1). Consequently, a vicious cycle with ROS-induced CaMKII-activation and subsequent cellular Ca^2+^ and Na^+^ overload with its deleterious effects on cardiac function develops (Figure 1).

## 3. Effects of SGLT2i on Cardiac Na^+^ Homeostasis 

There is overwhelming evidence that SGLT2i are cardioprotective in diabetic and non-diabetic patients. Consequently, a vast number of trials were conducted to decipher the underlying mechanisms. While the proposed mechanisms are multifaceted, a striking number of observations are directly or indirectly linked to effects on cellular Na^+^ handling, as set out below (Figure 2).

### 3.1. Inhibition of NHE1

There are several studies showing that inhibition of NHE1 (usually by its specific inhibitor cariporide) prevents development or worsening of heart failure [60,61,84,85,86,87]. The first work assuming an effect of SGLT2i on NHE1 was by Baartscheer et al. who observed that treatment of isolated ventricular cardiomyocytes of healthy mice and rabbits with empagliflozin reduced [Na^+^]_i_ as well as systolic and diastolic [Ca^2+^]_i_ probably due to decreased NHE1 activity [13]. In addition [Ca^2+^]_m_, assessed by the use of a mitochondrially targeted fluorescence resonance energy transfer (FRET-)-based Ca^2+^ indicator (mitoCam), increased [13]. In a later work the same group could reproduce NHE1 inhibition by dapagliflozin and canagliflozin in healthy murine cardiomyocytes and suggested direct binding of SGLT2i to the Na^+^ binding site of NHE1 by applying molecular docking studies [14]. Furthermore, we could recently show inhibition of NHE1-dependent pH recovery also in isolated human atrial cardiomyocytes, suggesting a potential role for the observed cardioprotection in clinical trials [15]. Interestingly, in a HFpEF model using Dahl salt-sensitive rats fed with high salt diet, chronic treatment with dapagliflozin (6 weeks) attenuated Ca^2+^ and Na^+^ overload and increased Ca^2+^ transient amplitude [88]. Furthermore, upregulation of NHE1, SGLT1, CaMKII, Na_v_1.5, and NCX1 in the diseased hearts was found to be attenuated by chronic dapagliflozin treatment. Importantly, the authors could not detect acute effects of dapagliflozin treatment on [Na^+^]_i_ or [Ca^2+^]_i_ in isolated myocytes. Unfortunately, NHE1 function in cardiomyocytes was not assessed in this study. However, inhibition of NHE1 activity by dapaglifozin was demonstrated in human umbilical vein endothelial cells (HUVEC) [88]. Noteworthy, Chung et al. could not detect NHE1 inhibition or a decrease in [Na^+^]_i_ by SGTL2i treatment neither in unpaced healthy human nor animal (mouse, rat and guinea pig) cardiomyocytes [89]. Interestingly, in patch clamp experiments they also assessed Na^+^/K^+^ ATPase current without differences between treatment groups. To address the conflicting data, Zuurbier et al. tested the influence of differences in the experimental setup (e.g., different pH-buffering systems, pacing of cells, extracellular pH, DMSO concentration) on the contradictory results and concluded that different conditions could not account for the discrepancies [90]. After transformation of the SBFI ratio-data of Chung et al. into [Na^+^]_i_ they even suggest that there is also a detectable decrease in [Na^+^]_i_ upon treatment with empagliflozin [90]. Overall, if NHE1 inhibition by SGLT2i can indeed convey a decrease in [Na^+^]_i_ in **healthy** cardiomyocytes is debatable, especially when considering that NHE1 activity at normal intracellular pH is low and contributes only little to cellular Na^+^ loading [91]. In a mouse model of ischemia-reperfusion (IR), empagliflozin delayed the time to onset of contracture as well as IR injury similar to cariporide, again pointing to a NHE1-dependent effect [92]. However, if a potential NHE1 inhibition by SGLT2i might contribute to a decrease in Na^+^ influx in **failing** cardiomyocytes with disturbed pH regulation [93] has not been systematically investigated to date.

### 3.2. Inhibition of Late I_Na_

Since late I_Na_ is a major contributor to Na^+^ overload in heart failure and inhibition of this current has proven to be cardioprotective in many models of cardiac disease [19,94,95], the idea of SGLT2i-mediated inhibition of this current seems appealing. Indeed, a recent study using mice with TAC (transverse aortic constriction)-induced heart failure as well as HEK293T cells transfected with Na_v_ 1.5 channels harboring LQT3 mutations (causing increased late I_Na_) showed that empagliflozin was able to significantly reduce late I_Na_ without affecting peak I_Na_ under the chosen experimental conditions (holding potential −120 mV, pacing frequency 1 Hz) [17]. This desirable preference for late I_Na_ over peak I_Na_ is shared by class Ib anti-arrhythmic drugs as well as ranolazine, but might—as for these drugs—become less pronounced under more physiological membrane potentials and at higher heart rates [96,97]. In a next step they observed that the incidence of spontaneous calcium transients evoked by treatment of healthy murine cardiomyocytes with the alkaloid veratridine, a specific stimulator of late I_Na_, was rapidly and reversibly reduced. Applying a homology-modeling approach based on the structure of human Na_v_1.4 Philippaert and colleagues developed a transmembrane model of human Na_v_1.5. Molecular docking simulations and introduction of targeted mutations revealed the amino acids F1760 and W1345 in the fDIII-DIV site as putative interaction partners with SGLT2i [17]. Intriguingly, F1760 was also reported to be elementary in the binding of local anesthetics (e.g., lidocaine) [97,98] as well as ranolazine [99]. However, apart from a direct interaction between SGLT2i and Na_v_1.5 channels, there is also the possibility of an indirect inhibition of late I_Na_ by interference with current regulation e.g., by CaMKII, a possibility, which was not investigated so far.

### 3.3. Inhibition of CaMKII

CaMKII is a serine/threonine kinase, which is markedly upregulated in the failing heart and contributes fundamentally to the development and progression of heart failure due to detrimental effects on cardiac excitation-contraction as well as excitation-transcription coupling [16,50,51,52,81,100]. Apart from the above mentioned augmentation of late I_Na_ by direct channel phosphorylation (see Section 2.2.1), CaMKII is able to phosphorylate ryanodine receptors (RyR2) at Serine 2814 thus increasing channel open probability with consequent spontaneous diastolic Ca^2+^ release events (i.e., Ca^2+^ sparks) that in turn may prompt further diastolic SR Ca^2+^ release of adjacent RyR2 clusters giving rise to diastolic Ca^2+^ waves [100]. Ca^2+^ sparks as well as waves result in an elevated diastolic [Ca^2+^]_i_ and activate NCX forward mode resulting in a depolarizing Na^+^ influx (I_ti_) that triggers delayed afterdepolarizations (DAD) [83] and contributes to cellular Na^+^ overload. Intriguingly, CaMKII is also known to stimulate NHE1 activity [64]. Thus, CaMKII inhibition could also be accountable for the observed inhibition of NHE1 activity upon SGLT2i treatment. 

Importantly, we could recently show that acute exposure of isolated healthy and failing (5 weeks after TAC) murine cardiomyocytes to empagliflozin at a clinically relevant dose (1 µmol/L) for 24 h reduced CaMKII activity as assessed by HDAC4 pulldown assays [10]. This was accompanied by a decreased CaMKII-dependent RyR2 phosphorylation at S2814 not only in murine but also in human failing cardiomyocytes. In line with this, empagliflozin treatment reduced Ca^2+^ spark frequency and increased SR Ca^2+^ load as well as Ca^2+^ transient amplitude, as a cellular surrogate for improved contractile function. Furthermore, measurement of SBFI ratios revealed a reduction in bulk [Na^+^]_i_ in healthy murine ventricular myocytes after 24 h of treatment. Interestingly, while the effects on CaMKII activity as well as Ca^2+^ transient amplitude were only observable after incubation with empagliflozin for 24 h, we could detect a decrease in subsarcolemmal [Na^+^] as assessed by application of the Nernst equation to the reversal potential of NCX currents, which were measured by whole cell patch clamp, as early as 30 min after start of treatment. Consistently, Pabel et al. reported no change in systolic Ca^2+^ transient amplitude, Ca^2+^ transient decay kinetics, or diastolic Ca^2+^ concentration in human failing cardiomyocytes upon acute empagliflozin exposure (15 min) [101]. The effects of prolonged SGLT2i treatment on Ca^2+^ handling were reproduced in a rat model of metabolic syndrome, where two weeks of dapagliflozin treatment also improved Ca^2+^ transient amplitude, SR Ca^2+^ load and increased left ventricular developed pressure [102]. Surprisingly, peak I_Na_ was reported to be significantly reduced in this study, while bulk [Na^+^]_i_ was not influenced by dapagliflozin [102]. In addition, the authors ascribed shortening of the prolonged QTc time and action potential duration to an increase in repolarizing K^+^ currents as well as the decrease in peak I_Na_. However, if a reduction of late I_Na_ might be involved, remained elusive. Overall, the beneficial effects were attributed to an improved mitochondrial function with decreased ROS production. Unfortunately, the final link between oxidative stress and the observed effects, which might e.g., be a decrease in CaMKII oxidation, was not investigated [102].

In summary, to date it is unclear, how SGLT2i interact with CaMKII activity. However, given the described time course of SGLT2i effects on CaMKII activity, Ca^2+^- and Na^+^ -handling it is possible that reduced CaMKII activity might be downstream of the direct SGLT2i target. Potential upstream targets could be late I_Na_, NHE1 and/or reduced oxidative stress.

### 3.4. Effect on Other Na^+^ Transporters (SGLT1 and SMIT1) 

In contrast to SGLT2, which is not present in the heart, two other members of the SLC5A gene family, namely SGLT1 (SLC5A1) and the sodium-myoinositol cotransporter-1 (SMIT1, SLC5A3) are expressed in the heart and have been shown to contribute to Na^+^ influx in cardiac disease [33,34,103,104]. In a mouse model of MI (LAD ligation) induced heart failure, SGLT1 expression was significantly upregulated and pretreatment with the specific inhibitor KGA-2727 attenuated left ventricular systolic dysfunction and fibrosis [105]. Importantly, except for sotagliflozin, which was designed as a dual SGLT1/SGLT2 inhibitor, empagliflozin (2680-fold), dapagliflozin (1242-fold), and canagliflozin (155-fold) exhibit an increased selectivity for SGLT2 over SGLT1 making a considerable contribution of SGLT1 inhibition in vivo unlikely [106].

On the other hand, myocardial overexpression of SMIT1 was shown to mediate hyperglycemia-induced NOX2 activation with consequent ROS production [11]. If this is also relevant in heart failure without concomitant hyperglycemia, is still unknown. Noteworthy, IC50 of empagliflozin for SMIT1 is 8.3 µmol/L, which is beyond the average plasma concentration of the drug [107]. 

Consequently, although conceivable, a relevant contribution of SGLT1 and SMIT1 to SGLT2i-mediated decrease in [Na^+^]_i_ seems unlikely, but might warrant further investigation.

## 4. Potential Role of Reduced [Na^+^]_i_ for Cardioprotection by SGLT2i

Keeping in mind that SGLT2i-mediated cardioprotection is primarily based on reduction in heart failure events (e.g., hospitalization for heart failure) rather than on vascular endpoints (myocardial infarction, stroke) we, in the following, focus on direct effects on cardiac function. 

In the light of the central role of Na^+^ dysregulation for heart failure and the growing number of studies reporting effects of SGLT2i on Na^+^ handling, it is tempting to speculate that restoration of Na^+^ homeostasis might be pivotal. However, which of the observed beneficial effects on cardiac function and structure can indeed be explained by improved Na^+^ handling? 

### 4.1. SGLT2i and Oxidative Stress 

Oxidative stress is a hallmark of cardiac disease including heart failure [108,109] and centrally involved in pathways promoting hypertrophy, fibrosis, cell death, as well as directly or indirectly (e.g., via oxidation of different kinases) involved in perturbation of cellular ion homeostasis [25,54,110]. Cytosolic Na^+^ overload results in an extrusion of Ca^2+^ from the mitochondria mediated by the NCLX. This mitochondrial Ca^2+^ deficit impairs Ca^2+^-induced stimulation of Krebs cycle dehydrogenases, which ultimately causes a shortage in the reducing equivalents NADH and NADPH [36]. Consequently, anti-oxidative enzymes such as glutathione peroxidase cannot be adequately regenerated. Importantly, as discussed above, oxidative stress can in turn cause cellular Na^+^ accumulation (e.g., by stimulation of late I_Na_) leading to a vicious cycle [25]. Thus, inhibiting Na^+^-influx by SGLT2i might mitigate oxidative stress as a central player in cardiac pathogenesis. 

First reports of anti-oxidative effects of SGLT2i stem from diabetic mouse and rat models, where treatment with ipragliflozin reduced plasma and liver levels of biomarkers of inflammation (IL-6, TNFα, CRP) and oxidative stress (thiobarbituric acid reactive substances and protein carbonyl) [111,112]. 

Since then, several studies also reported anti-oxidative properties of SGLT2i in the heart. Intriguingly, in human ventricular cardiomyocytes isolated from HFpEF patients acute treatment with empagliflozin (60 min) significantly reduced markers of oxidative stress (H_2_O_2_, GSH, lipid peroxidation, and 3-nitrotyrosine) as well as inflammatory markers (ICAM-1, VCAM-1, TNFα, and IL-6) [113]. Furthermore, in a diabetic mouse model 8 weeks of empagliflozin treatment resulted in a decrease in oxidative stress by activation of Nrf2/ARE signaling and downregulation of NADPH-oxidase 4 (NOX4) [114]. Of note, in non-diabetic rats with left ventricular dysfunction after MI, treatment with empagliflozin for 2 weeks after surgery again decreased myocardial oxidative stress but also increased pyruvate dehydrogenase (PDH) activity and ATP levels, improved left ventricular systolic function, reduced myocardial fibrosis and hypertrophy [115]. In another interesting study by Durak et al. using metabolic syndrome rats dapagliflozin not only reduced oxidative stress by improving mitochondrial membrane potential as well as mitochondrial fusion-fission and increased ATP/ADP ratio, but also shortened QTc time, reduced peak I_Na_, and improved cellular Ca^2+^ handling (e.g., increased Ca^2+^ transients, increased SR Ca^2+^ load) [102]. Although none of these studies provided a direct link between the reduction of oxidative stress and a decrease in [Na^+^]_i_, it is conceivable that restoration of Na^+^ homeostasis contributes to these observations (Figure 3). However, further studies on the interplay between SGLT2i, [Na^+^]_i_, mitochondrial function and generation of ROS in heart failure are warranted.

### 4.2. SGLT2i and Contractile Function

When dealing with heart failure, it is important to differentiate between HFrEF and HFpEF. While patients with HFpEF display a normal systolic function but impaired diastolic function, HFrEF is defined by a decrease in cardiac systolic contractility (in addition to diastolic dysfunction) and the severity of contractile dysfunction is positively correlated with the occurrence of major adverse cardiac events including hospitalizations for heart failure as well as cardiovascular death [116,117]. Consequently, improving systolic contractile function (without increasing cardiac energy demand) in these patients, e.g., through ameliorated Na^+^ homeostasis, might be a key to improving cardiovascular outcomes [118,119,120]. 

There are several studies reporting an increase in LVEF upon SGLT2i treatment. In a small randomized, double-blind, and placebo-controlled clinical trial (*n* = 84) Santos-Gallego et al. reported an improvement in LVEF (6.0 ± 4.2 vs. −0.1 ± 3.9; *p* < 0.001) as well as a decrease in LV dimensions in non-diabetic HFrEF patients within 6 months of treatment with dapagliflozin [121]. A similar increase in LVEF was observed in a retrospective analysis of type II diabetic patients with HFrEF 24 months after initiation of SGLT2i treatment [122]. In a rat model of myocardial infarction (MI)-induced heart failure empagliflozin (start of administration 4 weeks after MI and continued for further 4 weeks) slightly improved contractile function and prevented renal insufficiency. Apart from normalization of increased SGLT2 expression these effects were primarily ascribed to inhibition of renal NHE3 [123]. Unfortunately, it remains unclear if the amelioration of cardiac dysfunction in this study of cardiorenal syndrome is due to improved renal function or due to direct cardiac effects [123]. Interestingly, in a non-diabetic mouse model of TAC-induced heart failure, empagliflozin treatment prevented worsening of LVEF assessed by echocardiography *in vivo* as well as *ex vivo* by an isolated perfused working heart system (excluding a role of extrinsic factors such as hemodynamics or nephroprotection) [124].

Basically, as mentioned above, improved contractility could, at least in part, be explained by the restoration of cardiac Na^+^-homeostasis with consequent beneficial effects on Ca^2+^ handling (Figure 3). However, only a few studies observing an improved contractile function in vivo also systematically investigated the underlying pathways. Proposed mechanisms include amelioration of adverse remodeling by activation of cardiac GTP enzyme cyclohydrolase 1 (cGCH1) with consequent activation of eNOS and nNOS resulting in an increase of NO levels and a decrease in O_2_^−^ and nitrotyrosine levels [125]. In another study, dapagliflozin was shown to improve cardiac remodeling and contractile function by inhibition of the mitogen-activated protein kinases (MAPK) JNK and p38 [126]. Interestingly, as MAPK are readily activated by ROS [127], oxidative stress might again play a central role.

### 4.3. SGLT2i and Diastolic Function

Even broader pre-clinical evidence than for improved systolic function exists for SGLT2i mediated attenuation of diastolic dysfunction. For instance, Tanaka et al. showed that in a cohort of 53 type II diabetic patients with heart failure (HFpEF, HFmrEF, and HFrEF) administration of dapagliflozin improved left ventricular longitudinal function assessed by global longitudinal strain (via speckle-tracking strain analysis) and left ventricular diastolic function (E/e’) within 6 months [128]. Similar beneficial effects on diastolic function in patients with type II diabetes mellitus were reproduced in smaller trials using empagliflozin or canagliflozin [129,130]. In another interesting randomized, double-blind, and placebo-controlled study Rau et al. demonstrated that empagliflozin improved diastolic function (E/e´) as early as one day after initiation in diabetic patients, while non-invasively assessed hemodynamic parameters (stroke volume index, cardiac index, vascular resistance index or pulse rate) were unchanged [131]. While the results from two large clinical trials investigating the effects of empagliflozin (EMPEROR-Preserved [132]) or dapagliflozin (DELIVER [133]) in reducing cardiovascular death and heart failure events in non-diabetic patients with HFpEF are awaited for late 2021 and early 2022, respectively, there is pre-clinical evidence for an SGLT2i-mediated attenuation of diastolic dysfunction irrespective of a diabetic condition. Connelly and colleagues used unilateral nephrectomy or implantation of DOCA (deoxycorticosterone acetate) pellets in combination with high-salt diet to induce hypertension with a consequent HFpEF phenotype [134]. In this model empagliflozin treatment for 5 weeks improved diastolic function and reduced myocardial hypertrophy without affecting fibrosis [134]. Recently, in a porcine model of MI-induced HFrEF, treatment with empagliflozin for 2 months was shown to improve diastolic function as assessed by transthoracic echocardiography, cardiac magnetic resonance imaging as well as invasive hemodynamics [135]. This was associated with reduced myocardial fibrosis, reduced oxidative stress as well as improved eNOS-NO-cGMP-PKG signaling with consequent increase in titin phosphorylation, which contributes—when hypophosphorylated—to cardiomyocyte stiffness and diastolic dysfunction [135,136]. In line with this, recent studies in rodent as well as human HFrEF and HFpEF demonstrated improved diastolic function upon empagliflozin treatment due to reduced passive myofilament stiffness, which was also explained by enhanced titin phosphorylation [101,113]. Again improved NO-sGC-cGMP-PKG signaling was observed, which was, at least in part, due to attenuation of oxidative pathways [101,113]. However, the reasons for decreased oxidative stress remain elusive as the observed downregulation of inflammatory pathways could also be secondary [113]. Of note, reduced [Na^+^]_i_ (e.g., due to decreased late I_Na_) as well as inhibition of CaMKII can contribute to both reduced oxidative stress and inhibition of inflammatory pathways (e.g., by inhibition of the NLRP3 inflammasome) [17] and might thus be centrally involved in SGLT2i-mediated amelioration of diastolic dysfunction (Figure 3).

### 4.4. SGLT2i and Arrhythmias

Heart failure patients are at an increased risk for atrial as well as ventricular arrhythmias. While there may be pronounced heterogeneities regarding the pathomechanisms of different types of arrhythmias, they all have one thing in common: arrhythmias occur when arrhythmic substrates (structural and/or electrical abnormalities, e.g., fibrosis) and triggers (early (EADs) and delayed afterdepolarizations (DADs)) coincide [137]. Typically, EADs result from prolongation of the APD allowing for the reactivation of LTCC. APD prolongation in turn is caused either by a decrease in repolarizing currents (i.e., outward K^+^ currents) and/or an increase in depolarizing currents (e.g., late I_Na_) [23,40,110,138]. On the other hand, DADs result from spontaneous diastolic Ca^2+^ release from the SR via RyR2, which activates a transient inward current that is mainly mediated by NCX. Spontaneous SR Ca^2+^ leak is either due to RyR2 dysfunction or an increased SR Ca^2+^ load. The latter one usually results from Na^+^ overload with consequent cellular Ca^2+^ overload [139,140]. Importantly, oxidative stress as well as increased CaMKII activity (partly due to oxidative activation) play a critical role in the underlying electrophysiological alterations [25,54,81,141]. Against this background it is tempting to speculate that SGLT2i, which were shown to reduce [Na^+^]_i_ as well as oxidative stress and inhibit CaMKII, also possess anti-arrhythmic effects (Figure 3). Apart from its potential effects on arrhythmic triggers SGLT2i have repeatedly been shown to also positively affect the arrhythmic substrate, especially by reduction of fibrosis, which can be explained by anti-inflammatory and anti-fibrotic effects that might also be mediated by restoration of intracellular Ca^2+^ and Na^+^ homeostasis [17,142]. In this regard it is important to mention that CaMKII activity in the diseased heart also regulates inflammation e.g., by stimulation of NFκB with consequent expression of complement factor B within cardiomyocytes [143,144,145].

Interestingly, in an explorative analysis of the DECLARE-TIMI 58 trial the incidence of atrial fibrillation (AF)/atrial flutter (AFL) in the dapagliflozin group was reduced by ~19% in this high-risk collective of type II diabetic patients independent of a history of heart failure, cardiovascular disease, or pre-existing AF [146]. In addition, in a huge cohort study with almost 160,000 diabetic patients the adjusted hazard ratio for new onset of arrhythmia (including supraventricular and ventricular arrhythmias) was 0.83 (CI 0.751–0.916) in the SGLT2i group [147]. Finally, a recently published meta-analysis including 22 trials with in sum 52,115 patients suggests that SGLT2i reduce the risk of AF, embolic stroke, and ventricular tachycardia (VT) with consistent associations for patients with diabetes mellitus, heart failure, or chronic kidney disease [148]. Thus, although evidence from prospective trials investigating the effects of SGLT2i on arrhythmic events as a primary outcome is still lacking, one can assume that SGLT2i have anti-arrhythmic properties.

This notion is further supported by a pre-clinical study. In a rat model of streptozotocin-induced type II diabetes 8 weeks of empagliflozin treatment reduced AF inducibility, reduced atrial interstitial fibrosis, and improved mitochondrial function [149]. These alterations were accompanied by an increased expression of PGC-1a, Nrf1, and Tfam, which are involved in mitochondrial biogenesis, as well as an increase in the expression of Mfn-1, OPA-1, and Drp-1, which are central to the regulation of mitochondrial fission/fusion [149].

Intriguingly, empagliflozin also reduced the occurrence of spontaneous calcium transients induced by the late I_Na_ activator veratridine in isolated healthy murine cardiomyocytes [17].

Overall, although anti-arrhythmic effects of SGLT2i seem plausible from our current understanding of the pharmacodynamic features of this class of drugs, further clinical and preclinical studies are warranted to corroborate this assumption and to clarify the potential underlying mechanisms.

## 5. Conclusions

In this narrative review we focused on the effects of SGLT2i on cardiomyocyte Na^+^ handling and its potential implications for the observed cardioprotective effects in vivo. 

Due to the great scientific interest in the identification of the cellular targets mediating the cardioprotective effects of SGLT2i a plethora of potential mechanisms of action was proposed in the recent years. As it is unlikely that one substance interacts with an array of pathway specific targets, it is tempting to speculate that SGLT2i positively influence maybe only a couple of targets that are far upstream in the pathological processes of cardiac disease in general and heart failure in particular. CaMKII overactivation, Na^+^- and Ca^2+^-overload, and oxidative stress are hallmarks of heart failure, tightly interrelated and common basis for features of cardiac disease such as contractile and diastolic dysfunction, cardiac hypertrophy, and fibrosis. Consequently, interference of SGLT2i with these pathways as one of maybe a few major mechanisms is an appealing thought. Indeed, as outlined in this review, a growing body of evidence has shown that SGLT2i directly or indirectly inhibit CaMKII, reduce oxidative stress, and restore [Na^+^]_i_ (potentially due to inhibition of late I_Na_, NHE1 and/or other Na^+^ -transporters (SGLT1, SMIT1)) and [Ca^2+^]_i_. However, in regard to these effects, it is still unclear what is chicken and what is egg. Does inhibition of CaMKII normalize ion homeostasis and decrease ROS levels or is it the other way around? To address this issue further research, e.g., using specific knock-out models, is required and will also aid to understand which of the observed beneficial effects of SGLT2i in vivo indeed rely on this common pathway.

## Figures and Tables

**Figure 1 ijms-22-07976-f001:**
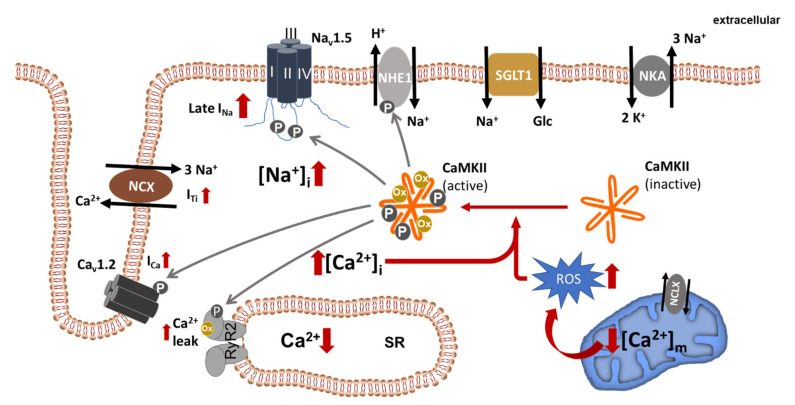
Dysregulated ion homeostasis in failing cardiomyocytes. Increased late I_Na_ as well as increased forward mode NCX (constituting I_Ti_), NHE1- and SGLT1 activity contribute to cellular Na^+^ overload ([Na^+^]_i_). Na^+^ overload, in turn, promotes cytosolic Ca^2+^ overload ([Ca^2+^]_i_) and mitochondrial Ca^2+^ depletion ([Ca^2+^]_m_) (with increased ROS generation), both of which are able to activate CaMKII, which is centrally involved in the regulation of multiple Na^+^ and Ca^2+^ channels/transporters. The solid grey lines indicate known CaMKII phosphorylation targets. NKA, Na^+^/K^+^ ATPase; RyR2, ryanodine receptor 2; SR, sarcoplasmic reticulum; NCX, Na^+^/Ca^2+^ exchanger; NCLX, mitochondrial Na^+^/Ca^2+^ exchanger; ROS, reactive oxygen species.

**Figure 2 ijms-22-07976-f002:**
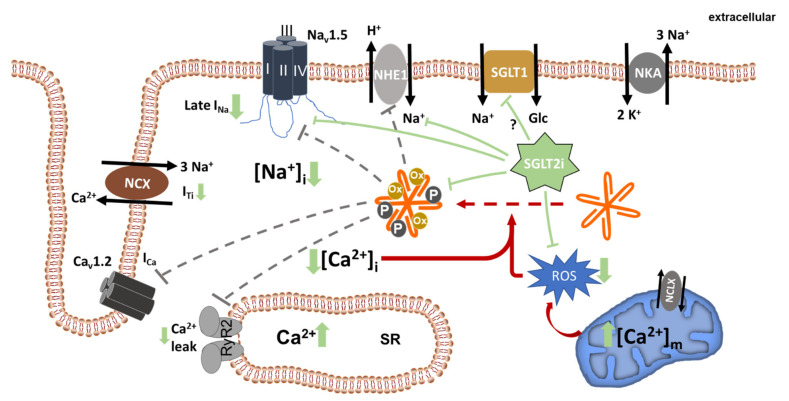
Beneficial effects of SGLT2i on cellular ion homeostasis. SGLT2i directly or indirectly inhibit NHE1, late I_Na_, and CaMKII, leading to a reduction in cellular Na^+^ and Ca^2+^ overload. As a consequence of reduced [Na^+^] and possibly also because of direct mitochondrial effects of SGLT2i, mitochondrial ROS generation is decreased. Dashed grey lines indicate potentially reduced interaction of CaMKII with its known targets. Solid green lines indicate inhibition mediated by SGLT2i. NKA, Na^+^/K^+^ ATPase; RyR2, ryanodine receptor 2; SR, sarcoplasmic reticulum; NCX, Na^+^/Ca^2+^ exchanger; NCLX, mitochondrial Na^+^/Ca^2+^ exchanger; ROS, reactive oxygen species.

**Figure 3 ijms-22-07976-f003:**
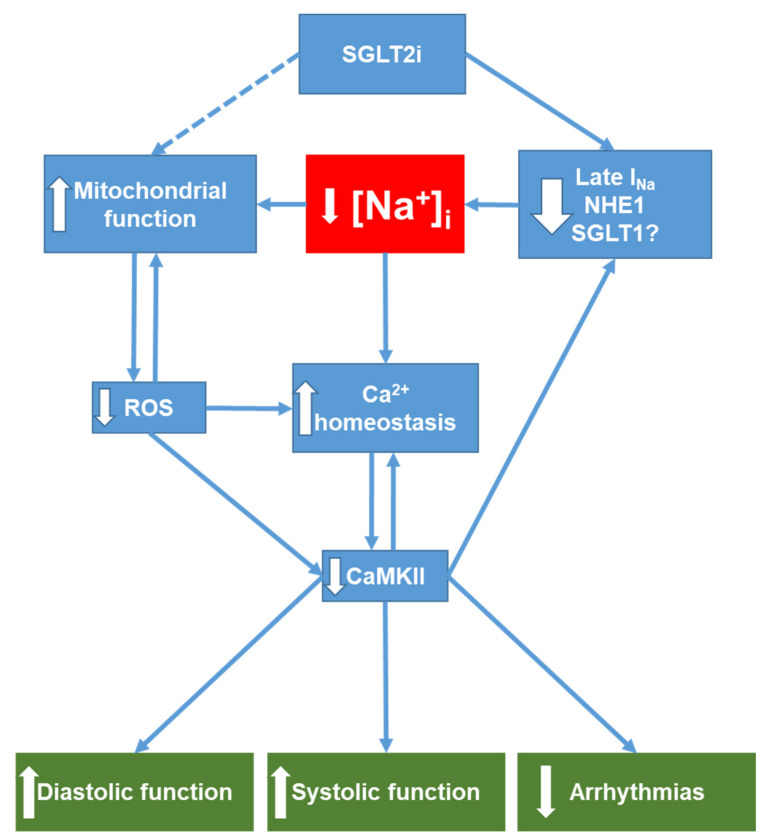
Overview of potential SGLT2i targets relevant for ion homeostasis in the heart. SGLT2i directly reduces (via inhibition of Late I_Na_, NHE1 and potentially SGLT1) intracellular Na^+^ accumulation in the failing heart. Decreased [Na^+^]_i_ in turn improves mitochondrial function (see Section 4.1) and thus may contribute to the decrease in mitochondrial ROS formation observed with SGLT2i treatment. Reduction of oxidative stress attenuates oxidative CaMKII activation, which, together with decreased [Na^+^]_i_, improves cellular Ca^2+^ handling and leads to improved diastolic and systolic function. Moreover, both restoration of Ca^2+^ homeostasis and CaMKII inhibition are antiarrhythmic. Downward-pointing arrows denote reduction or inhibition of a signaling pathway. Upward-pointing arrows denote stimulation or enhancement of a pathway. NHE1, Na^+^/H^+^ exchanger 1; SGLT1, sodium-glucose co-transporter 1; ROS, reactive oxygen species; CaMKII, Ca^2+^/calmodulin-dependent protein kinase II; [Na^+^]_i_, cytosolic Na^+^ concentration.

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
