# Peer review of "Cardioprotection by SGLT2 Inhibitors—Does It All Come Down to Na^+^?"

_ijms, 2021, doi:10.3390/ijms22157976_

Round 1

Reviewer 1 Report

Your article is a comprehensive review of current evidence on this topic. Some recommendations are provided. 

  1. In the abstract,  you could add a brief description of the relevant targets of SGLT2i on the mechanism of intracellular sodium homeostasis in failing myocardium.
  2. In the third section of this article, SGLT2i effects on contractile, diastolic function, and arrhythmogenesis and their connections with cardiomyocyte Na handling, could be depicted clearly and visualized in a summarized picture. 
  3. In this article, targets of cardiomyocyte intracellular sodium handling were lengthly described in the first and second sections. The whole picture of SGLT2i targets and their presumptive downstream pathways maybe not easily readable and impress your readers with 1-3 key points. Add a paragraph of the summarized mechanisms involved in cardioprotective effects of SGLT2i would be better. 

Author Response

Reviewer 1:

Comments and Suggestions for Authors

Your article is a comprehensive review of current evidence on this topic. Some recommendations are provided. 

1. In the abstract, you could add a brief description of the relevant targets of SGLT2i on the mechanism of intracellular sodium homeostasis in failing myocardium.

Answer: We thank the reviewer for this good advice. We added information on the most relevant targets mediating SGLT2i-induced changes in cellular Na+ handling in the abstract. It now reads:

 “On this basis we discuss the salutary effects of SGLT2i on Na+ homeostasis by influencing NHE1 activity, late INa as well as CaMKII activity. Finally, we highlight the potential relevance of these effects for systolic and diastolic dysfunction as well as arrhythmogenesis.”

2. In the third section of this article, SGLT2i effects on contractile, diastolic function, and arrhythmogenesis and their connections with cardiomyocyte Na handling, could be depicted clearly and visualized in a summarized picture. 

Answer: We thank the reviewer for this idea to improve our article. We now include an additional figure (Figure 3, see attached PDF) that summarizes the most relevant SGLT2i targets involved in cardiac ion homeostasis and clearly visualizes the links with cardiac function and arrhythmogenesis.

3. In this article, targets of cardiomyocyte intracellular sodium handling were lengthly described in the first and second sections. The whole picture of SGLT2i targets and their presumptive downstream pathways maybe not easily readable and impress your readers with 1-3 key points. Add a paragraph of the summarized mechanisms involved in cardioprotective effects of SGLT2i would be better. 

Answer: We hope the reviewer agrees that figure 3 can replace the suggested additional paragraph by visualizing the key points and shortly explaining them in the figure legend. Moreover, a brief summary of the key points is already given in the “conclusion” of our article.

Reviewer 2 Report

I have read this interesting and well-written article with great interest. The topic is very hot. The figures speak for themselves. This review is sure to be useful for readers.

This reviewer just raises a few suggestions.

1- In Introduction the authors stated "Importantly, in recent years several large clinical trials provided robust evidence that this class of drugs reduces cardiovascular death and heart failure events in patients with and without diabetes. These cardioprotective effects cannot be explained by an improved glycemic control." Actually, this statement is not entirely true, at least for MACEs, particularly ischemic events. In fact, the favorable impact that tight glycemic control can have in the course of ACS is well documented (Journal of Clinical Endocrinology and Metabolism Volume 97, Issue 3, March 2012, 933-942. doi: 10.1210/jc.2011-2037 - Journal of Diabetes Research, 2018, art. no. 3106056. doi: 10.1155/2018/3106056). In T2D, good glycemic control is achievable with SGLT2i. Therefore, the cardioprotective effect of these drugs is also mediated by a good glycemic control. This issue should be addressed by the authors, adding it in the introduction.

2- In a review, the references shouldn't be too old. Therefore, references older than 20 years should be deleted, or replaced with more recent ones.

Author Response

Reviewer 2:

I have read this interesting and well-written article with great interest. The topic is very hot. The figures speak for themselves. This review is sure to be useful for readers.

This reviewer just raises a few suggestions.

1- In Introduction the authors stated "Importantly, in recent years several large clinical trials provided robust evidence that this class of drugs reduces cardiovascular death and heart failure events in patients with and without diabetes. These cardioprotective effects cannot be explained by an improved glycemic control." Actually, this statement is not entirely true, at least for MACEs, particularly ischemic events. In fact, the favorable impact that tight glycemic control can have in the course of ACS is well documented (Journal of Clinical Endocrinology and Metabolism Volume 97, Issue 3, March 2012, 933-942. doi: 10.1210/jc.2011-2037 - Journal of Diabetes Research, 2018, art. no. 3106056. doi: 10.1155/2018/3106056). In T2D, good glycemic control is achievable with SGLT2i. Therefore, the cardioprotective effect of these drugs is also mediated by a good glycemic control. This issue should be addressed by the authors, adding it in the introduction.

Answer: The reviewer is correct that, in general, improved glycemic control has a beneficial effect on MACE. However, in EMPA-REG OUTCOME and DECLARE-TIMI 58, neither empagliflozin nor dapagliflozin significantly reduced the risk for the ischemic end points of myocardial infarction or stroke. Furthermore, a mediation analysis of EMPA-REG OUTCOME data (Inzucchi et al.) demonstrated that empagliflozin-mediated reduction in HbA1c contributed very little (~ 3%) to the observed beneficial effects. A similar mediation analysis from DECLARE-TIMI-58 was presented at the 2021 ACC Congress and confirmed the results of Inzucchi et al. (Berg et al, ACC congress 2021). This has led to the current European Society of Cardiology guidelines recommending initiation of therapy in diabetic patients with manifest coronary artery disease regardless of baseline HBA1c. In accordance, these data are supported by recent studies in patients with HFrEF, in whom SGLT2i prevented hospitalization for heart failure and cardiovascular death in diabetic and nondiabetic patients alike. For the revised manuscript, we added the following section to the introduction:

“Although tight glycemic control, which can be achieved by the addition of SGLT2i to the antidiabetic treatment regimen, has the potential to positively affect major adverse cardiovascular events (MACE) [2, 3], the risk of myocardial infarction or stroke were not significantly reduced by treatment with SGLT2i [4–6]. Moreover, a mediation analysis of EMPA-REG OUTCOME by Inzucchi et al. demonstrated that the reduction in HbA1c levels by empagliflozin contributed only slightly (~ 3%) to the observed cardioprotective effects [7]. These data are corroborated by recent studies in patients with heart failure, in whom SGLT2i prevented hospitalization for heart failure and cardiovascular death in diabetic and nondiabetic patients alike [8, 9].”

2- In a review, the references shouldn't be too old. Therefore, references older than 20 years should be deleted, or replaced with more recent ones.

Answer: We agree with the reviewer that a current review should not include outdated references. For this reason, we thoroughly reviewed the references again and decided to replace the following reference:

Le Prigent, K.; Lagadic-Gossmann, D.; Feuvray, D. Modulation by pH0 and intracellular Ca2+ of Na(+)-H+ exchange in dia-betic rat isolated ventricular myocytes. Circulation research, 1997, 80, 253–260.

Replaced by:

Vila-Petroff, M.; Mundiña-Weilenmann, C.; Lezcano, N.; Snabaitis, A.K.; Huergo, M.A.; Valverde, C.A.; Avkiran, M.; Mattiazzi, A. Ca(2+)/calmodulin-dependent protein kinase II contributes to intracellular pH recovery from acidosis via Na(+)/H(+) exchanger activation. Journal of molecular and cellular cardiology, 2010, 49, 106–112

Furthermore, we deleted the following references:

Snabaitis, A.K.; Yokoyama, H.; Avkiran, M. Roles of mitogen-activated protein kinases and protein kinase C in al-pha(1A)-adrenoceptor-mediated stimulation of the sarcolemmal Na(+)-H(+) exchanger. Circulation research, 2000, 86, 214–220.

Fliegel, L.; Walsh, M.P.; Singh, D.; Wong, C.; Barr, A. Phosphorylation of the C-terminal domain of the Na+/H+ exchanger by Ca2+/calmodulin-dependent protein kinase II. The Biochemical journal, 1992, 282 (Pt 1), 139–145.

Kim, C.H.; Fan, T.H.; Kelly, P.F.; Himura, Y.; Delehanty, J.M.; Hang, C.L.; Liang, C.S. Isoform-specific regulation of myocar-dial Na,K-ATPase alpha-subunit in congestive heart failure. Role of norepinephrine. Circulation, 1994, 89, 313–320.

Leem, C.H.; Lagadic-Gossmann, D.; Vaughan-Jones, R.D. Characterization of intracellular pH regulation in the guinea-pig ventricular myocyte. The Journal of physiology, 1999, 517 (Pt 1), 159–180.

However, we chose to retain key references that mark the first description of novel mechanisms or pathways or uniquely explore a particular topic (not comparably replicated in a more recent study); e.g.  Undrovinas, A.I.; Fleidervish, I.A.; Makielski, J.C. (Inward sodium current at resting potentials in single cardiac myocytes induced by the ischemic metabolite lysophosphatidylcholine. Circulation research, 1992, 71, 1231–1241): the first description of late INa.

We hope the reviewer agrees.